# Psychologic stress and disease activity in patients with inflammatory bowel disease: A multicenter cross-sectional study

**Manabu Araki[1], Shinichiro Shinzaki[1], Takuya Yamada[2], Shoko Arimitsu[3], Masato Komori[4], Narihiro Shibukawa[5], Akira Mukai[6], Sachiko Nakajima[7], Kazuo Kinoshita[8], Shinji Kitamura[9], Yoko Murayama[10], Hiroyuki Ogawa[11], Yuichi Yasunaga[12], Masahide Oshita[13], Hiroyuki Fukui[14], Eiji Masuda[15], Masahiko Tsujii[16], Shoichiro Kawai[1], Satoshi Hiyama[1], Takahiro Inoue[1], Hitoshi Tanimukai[17], Hideki Iijima[1]\*, Tetsuo Takehara[1]**

1 Department of Gastroenterology and Hepatology, Osaka University Graduate School of Medicine, Suita, Osaka, Japan, 2 Department of Gastroenterology, National Hospital Organization Osaka National Hospital, Osaka, Japan, 3 Kinshukai Infusion Clinic, Osaka, Japan, 4 Department of Gastroenterology, Osaka Rosai Hospital, Sakai, Osaka, Japan, 5 Department of Gastroenterology, NTT-West Osaka Hospital, Osaka, Japan, 6 Department of Gastroenterology, Sumitomo Hospital, Osaka, Japan, 7 Department of Gastroenterology, Toyonaka Municipal Hospital, Toyonaka, Osaka, Japan, 8 Department of Gastroenterology, Otemae Hospital, Osaka, Japan, 9 Department of Gastroenterology, Sakai City Medical Center, Sakai, Osaka, Japan, 10 Department of Gastroenterology and Hepatology, Itami City Hospital, Itami, Hyogo, Japan, 11 Department of Gastroenterology, Nishinomiya Municipal Central Hospital, Nishinomiya, Hyogo, Japan, 12 Department of Gastroenterology, Hyogo Prefectural Nishinomiya Hospital, Nishinomiya, Hyogo, Japan, 13 Department of Internal Medicine, Osaka Police Hospital, Osaka, Japan, 14 Department of Gastroenterology, Yao Municipal Hospital, Yao, Osaka, Japan, 15 Department of Gastroenterology, National Hospital Organization Osaka-minami National Hospital, Kawachinagano, Osaka, Japan, 16 Department of Gastroenterology, Higashiosaka City Medical Center, Higashiosaka, Osaka, Japan, 17 Faculty of Human Health Sciences, Graduate School of Medicine, Kyoto University, Kyoto, Japan

\* hiijima@gh.med.osaka-u.ac.jp

## Abstract

### Background and aims

Psychologic stress can affect the pathogenesis of inflammatory bowel disease (IBD), but the precise contribution of psychologic stress to IBD remains unclear. We investigated the association of psychologic stress with disease activity in patients with IBD, especially in terms of mental state and sleep condition.

### Methods

This was a multi-center observational study comprising 20 institutions. Data were collected using survey forms for doctors and questionnaires for patients, and the association of psychologic stress with clinical parameters was investigated. Mental state was evaluated using the Center for Epidemiologic Studies Depression (CES-D) scale, and sleep condition was evaluated by querying patients about the severity of insomnia symptoms.

### Results

A total of 1078 IBD patients were enrolled, including 303 patients with Crohn's disease and 775 patients with ulcerative colitis. Seventy-five percent of IBD patients believed that

**Funding:** This work was supported by a Grant-in-Aid from the Japan Society for the Promotion of Science (Grant No. 26460969). The funder did not have any additional role in the study design, data collection and analysis, decision to publish, or preparation of the manuscript.

**Competing interests:** The author has declared that no competing interests exist. The authors have no competing interests which can alter our adherence to PLOS ONE policies on sharing data and materials.

psychologic stress triggered an exacerbation of their disease (PSTE group) and 25% did not (non-PSTE group). The CES-D scores were significantly higher for patients with clinically active disease than for those in remission in the PSTE group (median (interquartile range) = 7 (4–9.5) vs. 5 (3–7), p < .0001), but not in the non-PSTE group (5 (2–8) vs. 4 (3–7), p = 0.78). Female sex and disease exacerbation by factors other than psychologic stress were independent factors of psychologic stress-triggered disease exacerbation. Also, patients with insomnia had higher disease activity than those without insomnia, especially in the PSTE group.

## Conclusions

A worsened mental state correlates with disease activity in IBD patients, especially those who believe that their disease is exacerbated by psychologic stress.

## Introduction

Inflammatory bowel disease (IBD), comprising Crohn's disease (CD) and ulcerative colitis (UC), is a refractory disease with repeated remission and exacerbation of symptoms. The etiology of IBD is unknown, but multiple factors may be associated with its pathogenesis. Several reports indicate that environmental factors, such as smoking, diet, infections, sleep, stress, and medication, affect the disease course by interacting with genetic susceptibility and impaired immune system responses. [1, 2] The sum of these environmental factors during the lifetime, termed the "exposome", appears to be associated with the pathogenesis of IBD. [3] Other environmental factors, such as psychologic stress and sleep condition, may also be associated with the IBD disease course: IBD patients are significantly more depressive than healthy controls, [4] and psychologic stress is associated with an exacerbation of IBD symptoms. [5] Several studies report impaired sleep quality in IBD patients [6–9] and the association of impaired sleep quality with an increased risk of relapse in patients in remission. [8, 10] In contrast, other reports demonstrated no association between psychologic stress and disease exacerbation, [11–14] and the precise etiology of insomnia in patients with IBD is not established. To date, mental status in patients with IBD has not been well assessed, which may account for the contradictory results. Moreover, some environmental factors, such as diet [15] and smoking [16], are known to affect disease activity in patients with IBD, but no studies have focused on psychologic stress and disease activity. In the present study, we aimed to clarify the association between psychologic stress and disease activity in a multicenter large cohort of IBD patients by analyzing their responses to detailed questionnaires about psychologic stress.

## Patients and methods

### Patients

This was a multicenter cross-sectional observational study conducted by the Osaka Gut Forum comprising 20 institutions in Japan. From November 2013 to August 2014, a total of 1078 IBD patients were enrolled in the present study, including 303 CD patients and 775 UC patients. All patients were Japanese and data were collected from survey forms provided to doctors and questionnaires provided to patients. The patient characteristics are shown in Table 1, and the number of patients included for each analysis in this study is shown in Fig 1. The study protocol was approved by the ethics committee of Osaka University Hospital, the ethics committee

of National Hospital Organization Osaka National Hospital, the ethics committee of Kinshukai Infusion Clinic, the ethics committee of Osaka Rosai Hospital, the ethics committee of NTT-West Osaka Hospital, the ethics committee of Sumitomo Hospital, the ethics committee of Toyonaka Municipal Hospital, the ethics committee of Otemae Hospital, the ethics committee of Sakai City Medical Center, the ethics committee of Itami City Hospital, the ethics committee of Nishinomiya Municipal Central Hospital, the ethics committee of Hyogo Prefectural Nishinomiya Hospital, the ethics committee of Osaka Police Hospital, the ethics committee of Yao Municipal Hospital, the ethics committee of National Hospital Organization Osaka-minami National Hospital, and the ethics committee of Higashiosaka City Medical Center. The written informed consent was waived by the ethics committees by giving the participants the opportunity to opt out.

## Medical records

The survey form queried the patients' background, including sex, disease duration, blood type, family history, smoking status, age at disease onset, surgical history for IBD, perianal disease, and extra-intestinal manifestations, and disease-specific items such as disease location and disease activity. Age at onset was categorized according to the Montreal classification [17] into two groups: A1/A2 (40 years old or younger) and A3 (over 40 years old). Disease location was also categorized according to the Montreal classification. [17]

## Patient questionnaire

Patients filled out the questionnaire regarding potential factors of disease exacerbation. We asked "What factor do you think is the cause of disease exacerbation? Please choose all of the alternatives that apply to you. (multiple answers allowed)", and patients selected from the following alternatives, "seasonality", "problem with work or family", "psychologic stress", "infections", "other than above", and "not sure" (S1 Fig). In addition to this information, there were questions regarding disease activity, mental state, and sleep condition during the past week.

**Table 1. Patient characteristics.**

|  | CD n = 303 | UC n = 775 |
|---|---|---|
| Age at enrollment, years, median (IQR) | 42 (32–50) | 48 (38–62) |
| Age at onset, years, median (IQR) | 24 (19–33) | 36 (24–50) |
| Sex, n, female/male | 84/219 | 371/402 |
| Family history, n, yes/no | 14/283 | 51/710 |
| Smoking, n, non/past or present | 195/104 | 543/214 |
| Montreal A, n, A1/A2/A3 | 26/231/35 | 34/424/295 |
| Montreal L, n, L1/L2/L3 | 96/45/157 | – |
| Montreal B, n, B1/B2/B3 | 92/110/63 | – |
| Proctitis/Left-sided colitis/Pancolitis, n | – | 187/226/343 |
| Perianal lesion, n, yes/no | 114/183 | 7 /759 |
| Extraintestinal manifestation, n, yes/no | 27/272 | 39/698 |
| Colitic cancer, n, yes/no | 5/297 | 7/765 |
| Surgery for IBD, n, yes/no | 153/137 | 22/674 |
| Disease activity, median (IQR) | 80.6 (34.7–139) (CDAI) | 1 (0–2) (partial Mayo score) |
| CES-D score, median (IQR) | 6 (3–8) | 5 (3–7) |
| Sleeping pill use, n, yes/no | 25/272 | 37/725 |
| Psychotropic drug use, n, yes/no | 8/279 | 14/748 |

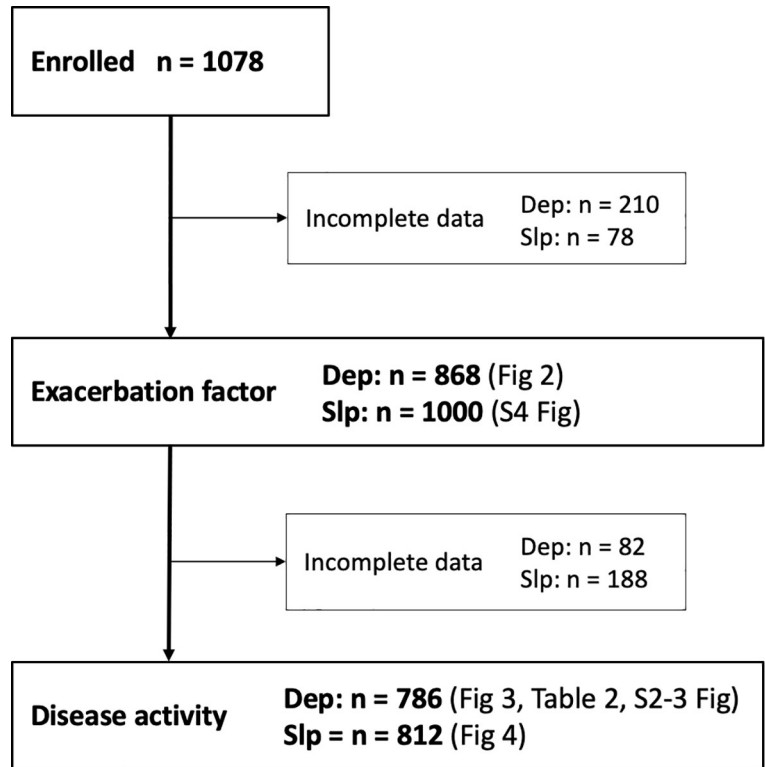

**Fig 1. Study flow chart.**

## Evaluation of disease activity

Disease activity in CD patients was evaluated according to the Crohn's disease activity index (CDAI) [18] and in UC patients according to the partial Mayo score. [19] Patients with a CDAI score of 150 or more and a partial Mayo score of 2 or more were defined as having active disease.

## Evaluation of mental state and sleep condition

To evaluate each patient's mental state, we used an 11-item form of the Center for Epidemiologic Studies Depression (CES-D) scale, Japanese version. [20, 21] The CES-D scale is a validated scale, comprising 9 "negative" items and 2 "positive" items, with each "negative" item scored on a Likert scale from 0 (none of the time) to 2 (all of the time) and each "positive" item scored from 0 (all of the time) to 2 (none of the time) (S1 Table). The total points range from 0 to 22 and a depressive state was defined as a CES-D score of at least 7. [22] To assess the general sleep condition, we used a simple version of a sleep questionnaire based on the validated Insomnia Severity Index (ISI) [23] to reduce recall bias. Patients were asked how much they felt their daily life was affected by their sleep condition during the past week, scored on a Likert scale from 1 (not at all) to 5 (exceedingly). We defined a Likert score of 3 or more as insomnia. [22, 24]

## Statistical analysis

Statistical analyses were performed using the Pearson's chi-square test and Mann-Whitney U test, and multivariate logistic regression analysis was performed with factors associated with disease exacerbation induced by psychologic stress. In logistic regression analysis, the covariates with significant difference in the univariate analysis were selected. To estimate $p$ values for interactions, two-way analysis of variance was used for continuous variables and Cox regression analysis for categorical data. All of the data obtained from this study were analyzed using JMP Pro version 12 and statistical significance was set at a $p$ value less than 0.05.

## Results

### Seventy-five percent of IBD patients recognize psychologic stress as a factor for disease exacerbation

We first investigated the factors for disease exacerbation in patients with IBD. Of the 1078 IBD patients initially enrolled, 210 were excluded due to incomplete information about exacerbating factors, and therefore a total of 868 patients were included in the analysis of the exacerbation factor (Fig 1). Analysis of all the questionnaires revealed that 75.1% of the IBD patients responded that they believed their disease activity was exacerbated by psychologic stress, and psychologic stress was the most common factor for exacerbation among the alternative responses (Fig 2). When patients were divided into 2 groups–patients who believed that psychologic stress triggered an exacerbation of their disease (PSTE group) and those who did not (non-PSTE group)–the proportions of CD patients and UC patients were 75.0% and 75.2%, respectively. These data indicate that, among both CD and UC patients, 75% of IBD patients believed that psychologic stress exacerbated their disease.

### Disease activity in IBD patients is exacerbated by an altered mental state

We next analyzed the association between mental state and disease activity in 786 patients (Fig 1). In all IBD patients, the CES-D scores were significantly higher in patients with active disease than for those in remission (S2 Fig). The CES-D scores at the time of the survey were significantly higher for patients with clinically active disease than for those in remission in the PSTE group, but not in the non-PSTE group, with a $p$ value for interaction of 0.066 (Fig 3). We further investigated the correlation between CES-D scores and disease activity using a scatter diagram, which revealed a positive correlation between CES-D scores and disease activity in both CD and UC patients in the PSTE group, but no correlation in the non-PSTE group (S3 Fig). These findings revealed a positive association between CES-D scores and disease activity among IBD patients who believed psychologic stress triggered an exacerbation of their disease.

### Factors associated with the psychologic stress for disease exacerbation

We analyzed the factors associated with the belief that psychologic stress is involved in disease exacerbation by comparing the patient characteristics between the PSTE and non-PSTE groups. Univariate analysis followed by multivariate analysis revealed that problems with work or family, seasonal disease exacerbation, infections, and diet, along with female sex were independent factors associated with psychologic stress-triggered disease exacerbation (Table 2). These findings suggest that patients who believed that psychologic stress triggered an exacerbation of their disease are predominantly female, and think that problems with work or family exacerbate the disease but other environmental factors do not.

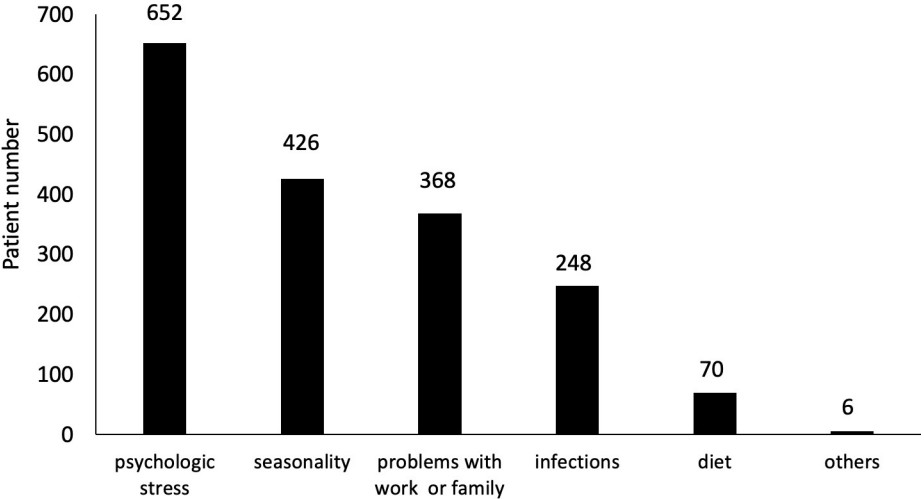

**Fig 2. Distribution of exacerbation factors.** Patients filled out questionnaires regarding possible disease exacerbation factors by selecting from alternative responses. Multiple answers were allowed.

## Insomnia is associated with active disease in IBD

We investigated the association between sleep condition and disease activity. A total of 1000 patients were included after excluding 78 patients due to incomplete answers regarding their insomnia symptoms (Fig 1). The proportion of total IBD patients with insomnia was 22.3%, with no significant difference between CD (25.7%) and UC (21.0%) patients. We also investigated the association between disease activity and sleep condition. The proportion of patients with active disease was significantly greater among patients with insomnia than among those without insomnia (S4 Fig). This difference was also observed in both CD and UC patients.

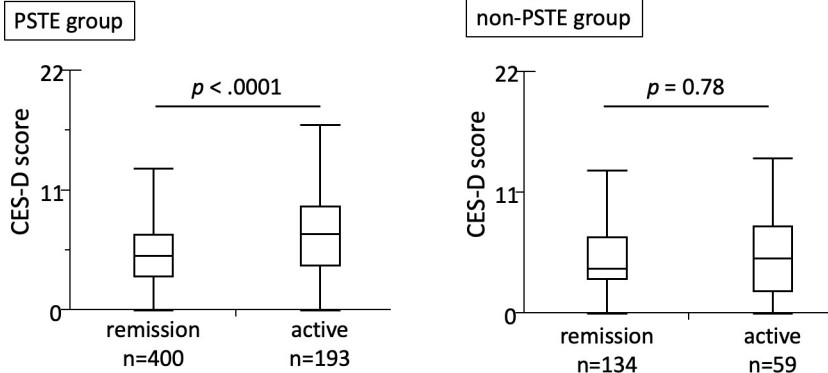

**Fig 3. Association between disease activity and depressive state in IBD patients.** Among patients who responded that they believed that psychologic stress triggered an exacerbation of their disease (PSTE group), the CES-D scores were significantly higher for patients with active disease than for those in remission (median (interquartile range (IQR)) = 7 (4–9.5) vs. 5 (3–7), $p < .0001$). Among patients who responded that they did not believe psychologic stress exacerbated their disease (non-PSTE group), the CES-D scores did not differ significantly between patients with active disease and patients in remission (median (IQR) = 5 (2–8) vs. 4 (3–7), $p = 0.78$).

**Table 2. Factors associated with psychologic stress-triggered disease exacerbation.**

| | Univariate analysis | | | Multivariate analysis | |
|---|---|---|---|---|---|
| | PSTE group | Non-PSTE group | *p* value | Odds ratio (95% CI) | *p* value |
| Sex (female), n (%) | 301 (46.2) | 82 (38.0) | 0.034 | 1.52 (1.07–2.18) | 0.021 |
| Exacerbation by problems with work or family (yes), n (%) | 305 (46.8) | 63 (29.2) | < .0001 | 1.97 (1.38–2.86) | 0.0002 |
| Seasonal disease exacerbation (yes), n (%) | 305 (53.5) | 121 (65.1) | 0.0058 | 0.62 (0.43–0.89) | 0.0091 |
| Exacerbation by infections (yes), n (%) | 165 (25.3) | 83 (38.4) | 0.0002 | 0.54 (0.37–0.77) | 0.0009 |
| Exacerbation by diet (yes), n (%) | 33 (5.1) | 37 (17.1) | < .0001 | 0.26 (0.15–0.45) | < .0001 |

### Insomnia is associated with disease activity especially in patients who believed that psychologic stress triggered their disease exacerbation

Finally, we investigated the association of insomnia and disease activity by analyzing the proportion of patients with insomnia between those with active disease and those in remission in the PSTE and non-PSTE groups. We excluded 188 patients due to incomplete responses regarding exacerbating factors for psychologic stress (Fig 1). In the PSTE group, the proportion of patients with insomnia was significantly higher for patients with active disease than for those in remission. No significant difference was detected in the non-PSTE group, with *p* value for the interaction of 0.437 (Fig 4). These data indicate that the association between insomnia and disease activity was particularly strong in the PSTE group.

## Discussion

The findings of the present study revealed that 75% of IBD patients considered their disease to be exacerbated by an altered mental state, and among these patients, disease activity correlated positively with CES-D scores, which is widely used as an indicator of the mental state. In addition to a previous report showing an association between psychologic stress and disease activity in IBD patients, [25] the present data provide new insight that a depressive state is associated with disease activity in a subgroup of IBD patients who believe that psychologic stress triggers an exacerbation of their disease. Reports on the association of psychologic status and insomnia with disease activity provide conflicting evidence, which might be due to the lack of patient stratification based on the patients' belief. Psychologic stress can influence gut

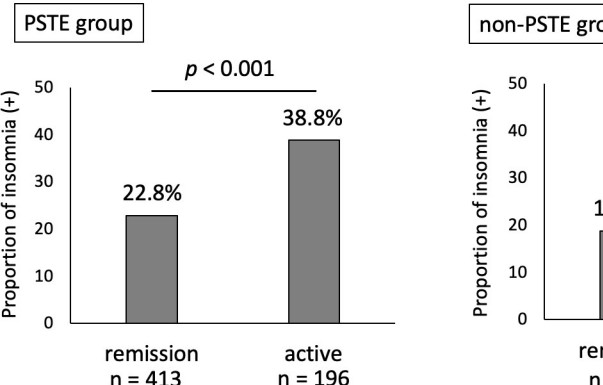

**Fig 4. Association between disease activity and insomnia in IBD patients.** In the PSTE group, the proportion of patients with insomnia (insomnia (+)) was significantly higher for patients with active disease than for those in remission (38.8% (76/196) vs. 22.8% (94/413), *p* < .0001). No significant difference was detected in the non-PSTE group (26.6% (17/64) vs. 18.7% (26/139), *p* = .2030, respectively).

inflammation through various mechanisms via the hypothalamus–pituitary–adrenal axis and the autonomic nervous system, resulting in the production of proinflammatory cytokines, activation of macrophages, and alteration of intestinal permeability and gut microbiota. [26–28] Also, psychotherapy by a counselor specially trained in the management of IBD improves the course of IBD in individuals with psychosocial stress, [29] indicating that managing psychologic stress may be a therapeutic target for IBD. Previous reports also show that female sex is associated with anxiety in IBD patients [30] as well as in the general population, [31] which is consistent with our present data that more female patients reported that psychologic stress induced disease exacerbation. Increased psychologic stress in the PSTE group may augment the fear for exacerbation and thereby affect these mechanisms, whereas other factor(s) may affect disease activity in the non-PSTE group. Moreover, we recently reported that seasonality of disease onset and exacerbation was observed especially in young-onset IBD patients. [32] Together, these data suggest that multiple environmental factors are associated with disease activity.

Insomnia was associated with the disease activity of IBD in the present study, consistent with previous studies. [6–8] Changes in the sleep architecture during acute infection correlate with the levels of plasma interleukin (IL)-6, [33] and IL-6 administration significantly decreases slow-wave sleep during the first half of sleep and increases slow-wave sleep during the second half. [34] Moreover, sleep deprivation increases the production of inflammatory cytokines, such as IL-6 and TNF-alpha. [35] In a murine colitis model, sleep deprivation worsened inflammation and delayed recovery. [36] In the present study, we showed that insomnia was associated with disease activity particularly in patients who believed that psychologic stress triggered an exacerbation of their disease. A previous study reported that depressive symptoms were a strong predictor of sleep disturbance, [10] and our present data also suggest that psychologic stress is associated with disease flare in patients with insomnia. The results of effect modification were different between depressive state and insomnia, indicating that psychologic stress-triggered disease exacerbation is strongly connected to the association between depressive state and disease activity, whereas the association between insomnia and disease activity cannot be ruled out, even in patients who do not recognize that psychologic stress triggers an exacerbation of their disease activity. Compared with insomnia, a depressive state might contribute more to increase disease activity in patients recognizing that psychologic stress triggers disease exacerbation, but the present study design does not allow for a direct comparison. To clarify this further, we are conducting a prospective study.

The present study has some limitations. First, because it was a cross-sectional observational study, we could not clarify whether a depressive state and insomnia are causes or consequences of exacerbated symptoms. To address this issue, we are now performing a prospective study to analyze disease activity, and the mental and sleep states at multiple time-points. Second, insomnia was assessed by self-reported questionnaires, not by objective measurements. Objective monitoring of sleep quality in a multicenter study setting, however, is associated with many difficulties, including equipment requirements, high cost, and adequate patient recruitment. Moreover, patient-reported outcomes have recently become an important aspect of assessing activity of IBD to optimize the therapeutic outcome and quality of life. [37, 38] The importance of patient-reported outcomes for assessing the clinical status of IBD, especially for monitoring the influence of environmental factors, should be discussed further.

In conclusion, our data revealed that a worsened mental state correlates with disease activity in IBD patients, especially those who believe that psychologic stress triggers an exacerbation of their disease. Although further studies are necessary to reveal the mechanism of the association between these environmental factors and the disease course, the present findings can be

applied to develop tailor-made therapies for IBD patients based on their levels of psychologic stress.

## Supporting information

**S1 Fig. The original version of the questionnaire.**
(TIFF)

**S2 Fig. Association between disease activity and depressive state in IBD patients.** In all IBD patients, the CES-D scores were significantly higher for patients with active disease than for those in remission (median (IQR) = 6 (4–9) vs. 5 (3–7), $p < .0001$).
(TIFF)

**S3 Fig. Scatter diagram to evaluate the correlation between the CES-D scores and disease activity among patients with CD (n = 177) and UC (n = 416) in the PSTE group and the non-PSTE group.** In the PSTE group, the CES-D scores correlated positively with disease activity (r = 0.26, p = 0.0004 in CD and r = 0.21, p = 0.0007 in UC). In patients with CD (n = 60) and UC (n = 133) in the non-PSTE group, the CES-D scores did not correlate with disease activity (r = -0.04, p = 0.98 in CD and r = 0.12, p = 0.17 in UC).
(TIFF)

**S4 Fig. Comparison of the proportion of patients with active disease in the insomnia (+) or insomnia (-) groups.** Among patients with CD and UC, the proportion of patients with active disease was significantly higher in the insomnia (+) group than in the insomnia (-) group.
(TIFF)

**S1 Table. The CES-D scale.**
(TIFF)

## Acknowledgments

We are grateful to Dr. E. Mita (National Hospital Organization Osaka National Hospital), Dr. H. Ito (Kinshukai Infusion Clinic), Dr. M. Yamamoto (Kinki Central Hospital of Mutual Aid Association of Public School Teachers), Dr. K. Tominaga (Kaizuka City Hospital), Dr. T. Kitada (Kawanishi City Hospital), Dr. Y. Okuda (Saiseikai Senri Hospital), and Dr. S. Ishii (Osaka General Medical Center) in the Osaka Gut Forum for their contributions to the data collection. We also thank Dr. T. Kitamura (Division of Environmental Medicine and Population Sciences, Department of Social and Environmental Medicine, Osaka University Graduate School of Medicine) for critical advice for statistical methods.

## Author Contributions

**Conceptualization:** Manabu Araki, Shinichiro Shinzaki, Hitoshi Tanimukai, Hideki Iijima, Tetsuo Takehara.

**Data curation:** Manabu Araki, Shinichiro Shinzaki.

**Formal analysis:** Manabu Araki, Shinichiro Shinzaki, Hideki Iijima.

**Funding acquisition:** Hideki Iijima.

**Investigation:** Shinichiro Shinzaki, Takuya Yamada, Shoko Arimitsu, Masato Komori, Narihiro Shibukawa, Akira Mukai, Sachiko Nakajima, Kazuo Kinoshita, Shinji Kitamura, Yoko Murayama, Hiroyuki Ogawa, Yuichi Yasunaga, Masahide Oshita, Hiroyuki Fukui, Eiji Masuda, Masahiko Tsujii, Shoichiro Kawai, Satoshi Hiyama, Takahiro Inoue.

**Methodology:** Shinichiro Shinzaki, Hitoshi Tanimukai.

**Supervision:** Hitoshi Tanimukai, Tetsuo Takehara.

**Writing – original draft:** Manabu Araki.

**Writing – review & editing:** Shinichiro Shinzaki, Hideki Iijima.

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
