## [Decision Letter · Decision Letter 0]

15 Oct 2019

PONE-D-19-26952

Vulnerability to psychologic stress and disease activity in patients with inflammatory bowel disease: A multicenter observational study

PLOS ONE

Dear Dr. Iijima,

Thank you for submitting your manuscript to PLOS ONE. After careful consideration, we feel that it has merit but does not fully meet PLOS ONE’s publication criteria as it currently stands. Therefore, we invite you to submit a revised version of the manuscript that addresses the points raised during the review process.

Although your manuscript is interesting, the two reviewers addressed several major and minor concerns about your manuscript. Please revise your manuscript carefully.

We would appreciate receiving your revised manuscript by Nov 29 2019 11:59PM. To enhance the reproducibility of your results, we recommend that if applicable you deposit your laboratory protocols in protocols.io, where a protocol can be assigned its own identifier (DOI) such that it can be cited independently in the future. For instructions see: http://journals.plos.org/plosone/s/submission-guidelines#loc-laboratory-protocols

We look forward to receiving your revised manuscript.

Kind regards,

Kenji Hashimoto, PhD

Academic Editor

PLOS ONE

Journal Requirements:

3. Please include additional information regarding the questionnaire regarding potential factors of disease exacerbation used in the study and ensure that you have provided sufficient details that others could replicate the analyses. For instance, if you developed this questionnaire as part of this study and it is not under a copyright more restrictive than CC-BY, please include a copy, in both the original language and English, as Supporting Information.

'This work was supported by a Grant-in-Aid from the Japan Society for the Promotion of Science (Grant No.

26460969).'

'The authors received no specific funding for this work.'

'The authors have declared that no competing interests exist.' 

We note that one or more of the authors are employed by a commercial company: Kinshukai Infusion clinic

6. Thank you for including your ethics statement: The ethics committees of all participating institutions approved the study protocol.

Additional Editor Comments (if provided):

Reviewers' comments:

Reviewer's Responses to Questions

**Comments to the Author**

1. Is the manuscript technically sound, and do the data support the conclusions?

Reviewer #1: Yes

Reviewer #2: Partly

2. Has the statistical analysis been performed appropriately and rigorously? 

Reviewer #1: Yes

Reviewer #2: Yes

3. Have the authors made all data underlying the findings in their manuscript fully available?

Reviewer #1: Yes

Reviewer #2: Yes

4. Is the manuscript presented in an intelligible fashion and written in standard English?

Reviewer #1: Yes

Reviewer #2: Yes

5. Review Comments to the Author

Reviewer #1: The authors conducted an interesting multicenter cross-sectional survey to evaluate the association between psychological factors and disease activity in 1078 patients with inflammatory bowel diseases (IBD). In this study, patients believed that psychological stress was the most common factor for a trigger of their disease exacerbation. The authors also explored the association between CES-D or insomnia and disease activity. In the exploratory analyses, the authors conducted subgroup analyses based on whether patients believe psychological stress is the most important factor for a trigger of their disease exacerbation.

The research question is interesting, and the data shown in this manuscript should be helpful for both healthcare providers and IBD patients to discuss the importance of assessing patients’ psychological condition. However, the manuscript is not straightforward. The authors should simply focus on their main research question in the manuscript. To make the manuscript simple and more reasonable, I have several comments below.

<major comments="">

1. The word, “vulnerability to psychological stress” may be improper because the authors did not measure the vulnerability such as lack of stress coping. I assume that “vulnerability to psychological stress” means that patients believed that psychological stress triggered their disease exacerbation in this study. This concept may be better to have a different name such as patients’ belief.

2. Instead of psychological stress, the authors measure depression (CES-D) and insomnia. The concept of psychological stress is different from depression and insomnia. The authors should clearly use the word such as depression and insomnia instead of psychological stress when they discuss depression and insomnia. In other words, the author should only use the word “psychological stress” when they discuss the results of questionnaire regarding factors related to disease exacerbation.

3. I believe that the most important information of this article may be that psychological stress was the most common factor for a trigger of their disease exacerbation. The authors should write the exact sentences to shown how they asked this question and what the choices were.

4. The authors used the word group 1 and group 2 which were based on whether patients answered whether psychological stress triggered their disease exacerbation. The expression of group 1 or group 2 may not be easy to follow for readers because name of group numbers (group 1 and group 2) are confusing. Please rename group1 and group 2 for readers’ understanding.

5. It seems that authors tried to show there is effect modification (i.e. interaction) between group 1 and group 2 regarding the association between CES-D or insomnia and disease activity. In the current analysis, the authors conducted only subgroup analyses based on group 1 and group2. Please show the p-value for interaction. Based on the p-value for interaction, the authors can discuss whether depression or insomnia are associated with disease activity in Group1 but not in Group 2.

6. The authors used CES-D as a continuous variable and showed the significant difference between remission and active patients in Group 1 in Figure 2. It may be better to use categorical variable such as depressive or not based on the cut-off written in the method section. If the authors use CES-D as a continuous variable, please discuss minimal clinically important difference (MCID) of CES-D.

7. Instead of table 1 and table 2, it may be better to show only patient characteristics in all patients, Group 1, and Group 2 in the same table without any statistical testing. In other words, the authors may delete table 1 and table 2, and they create a new table showing the patient characteristics.

8. I believe that the supplementary figure 1 is important because it shows patients’ selection. Please transfer the supplementary figure 1 to figure 1 in the manuscript.

9. There may be too many figures and supplementary figures. Please decrease the number of figures only focus on exacerbation factors and the subgroup analysis of Group 1/2 regarding depression and insomnia because this may be a main research question of this study. If this is not, the authors should show what is their main research question.

10. As shown in the supplementary figure 1, the authors selected the study population in two ways. If the authors decrease the number of figures and analyses, it may be possible to select patients in one process.

11. In the overall analyses, the authors showed only p-values in figures. Please show the point estimates such as mean difference and 95% intervals. This is important to discuss whether the difference is clinically important or not. This issue is related to MCID as I commented in No 6.

12. The authors should carefully interpret results in this study. Because this is the cross-sectional study, it is reasonable to consider the depression or insomnia is due to disease activity rather than vice-versa. Please discuss the results from the viewpoint above. Moreover, the authors should explain how patients’ belief (psychological stress triggered their disease exacerbation) affects the association of depression or insomnia with disease activity. In other word, please explain the mechanism of this effect modification if the authors find the significant interaction (please see the comments No 5).

13. Overall, there may be better words and sentences to show what the authors want to discuss in this manuscript. Please ask English editing to native and academic English editors.

<minor comments="">

1. It is better to show the study design in detail in the title. This is a multicenter cross-sectional survey or multicenter cross-sectional study.

2. In the introduction, the authors do not need to discuss seasonality because this is different from the current study purpose.

3. In the method section, please briefly mention the validity and reliability of CES-D and insomnia questionnaire based on previous questionnaire development study conducted by the developers.

4. In the introduction and discussion, the author should summarize previous studies evaluating patients’ belief of disease exacerbation triggers and discuss the discrepancy between the results of current study and those of previous studies with possible reasons.

5. In the introduction and discussion, the author should summarize previous studies evaluating depression or insomnia and disease activity, and please discuss the discrepancy between the results of current study and those of previous studies with possible reasons.

Reviewer #2: This cross sectional study assessed psychological stress and insomnia in approximately 1,000 IBD subjects. The main findings were that most patients believed that stress is associated with disease activity and, in patients who believe this, that insomnia was also associated with disease activity. The data are well presented and my main concern is with the discussion surrounding the data. It has been known for many years that most patients believe that disease activity not only results stress, anxiety and depression, but that the converse is also true (that stress, anxiety and depression results in subsequent disease activity) and the findings from this paper are not novel in any way. The authors mention in the discussion that they are performing a prospective study and this would be a much more interesting study, allowing them to determine causative associations rather than just associations. Indeed, Gracie et al. (Gastroenterology 2018) have already shown that there is a bi-directional association between psychological disability and disease activity. With regard to insomnia, it is well known that sleep disturbance is intimately associated with psychological disability while it is also known that sleep disturbance is particularly common in those with gastrointestinal disorders. Indeed it would be surprising if disordered sleep patterns were not associated with active disease, merely by the effect of abdominal pain and diarrhoea on sleep.

This does not mean that the study is of no value, but the discussion section should be altered to more accurately reflect the cross-sectional nature of the data. As an example, the statement " .. and the results suggest that psychological stress can be an initiator of disease" (line 281) is simply not appropriate. Similarly, the statement "our present data also suggest that high vulnerability to psychologic stress is associated with disease flare in patients with insomnia" (line 273) is not supported by the data. Finally, their conclusion "In conclusion, our data revealed that psychologic stress and insomnia affect disease activity in IBD patients, especially..." (line 292) is wrong, since it is just as likely that disease activity affects psychological stress and insomnia.

The English, references and figures are appropriate

 </minor></major>

6. PLOS authors have the option to publish the peer review history of their article (what does this mean?). If published, this will include your full peer review and any attached files.

Reviewer #1: No

Reviewer #2: No

---

## [Author Response · Author response to Decision Letter 0]

29 Mar 2020

Response to Reviewer #1: 

Thank you very much for reviewing our manuscript. We have revised the manuscript in response to all comments received.

1. The word, “vulnerability to psychological stress” may be improper because the authors did not measure the vulnerability such as lack of stress coping. I assume that “vulnerability to psychological stress” means that patients believed that psychological stress triggered their disease exacerbation in this study. This concept may be better to have a different name such as patients’ belief.

<Answer> Thank you for this suggestion. As the reviewer pointed out, we intended to investigate patients who believed that psychologic stress triggered an exacerbation of their disease. In the revised manuscript, we changed the title and amended the related sentences explaining the concept of the patients’ belief.

2. Instead of psychological stress, the authors measure depression (CES-D) and insomnia. The concept of psychological stress is different from depression and insomnia. The authors should clearly use the word such as depression and insomnia instead of psychological stress when they discuss depression and insomnia. In other words, the author should only use the word “psychological stress” when they discuss the results of questionnaire regarding factors related to disease exacerbation.

<Answer> To avoid misunderstanding, we used the phrase “psychologic stress” only when discussing the results of the questionnaire, and used the phrase “depressive state” when referring to the association between CES-D scores and disease activity.

3. I believe that the most important information of this article may be that psychological stress was the most common factor for a trigger of their disease exacerbation. The authors should write the exact sentences to shown how they asked this question and what the choices were.

<Answer> The exact sentences of the question are now provided on lines 143-147 in the revised manuscript. And according to the suggestion by the Journal Requirements, we have also included the original Japanese version as S1 Fig in Supporting Information.

4. The authors used the word group 1 and group 2 which were based on whether patients answered whether psychological stress triggered their disease exacerbation. The expression of group 1 or group 2 may not be easy to follow for readers because name of group numbers (group 1 and group 2) are confusing. Please rename group1 and group 2 for readers’ understanding.

<Answer> We renamed group 1 and group 2 as the PSTE (psychologic stress-triggered exacerbation) group and the non-PSTE group. Thank you for this suggestion. 

5. It seems that authors tried to show there is effect modification (i.e. interaction) between group 1 and group 2 regarding the association between CES-D or insomnia and disease activity. In the current analysis, the authors conducted only subgroup analyses based on group 1 and group2. Please show the p-value for interaction. Based on the p-value for interaction, the authors can discuss whether depression or insomnia are associated with disease activity in Group1 but not in Group 2.

<Answer> Figure 2 (new Fig 3), which shows the association between the CES-D scores and disease activity in each group, indicates a tendency toward an interaction with a p-value of 0.066. This result indicates a positive association between CES-D scores and disease activity only in the PSTE group and no association was observed in the non-PSTE group. As for Figure 5 (new Fig 4), which shows the association between the clinical activity scores and insomnia in each group, the p-values for the group interactions were 0.61 for CD and 0.23 for UC, respectively. These data indicate a positive association between insomnia and disease activity, especially in the PSTE group. Although significant differences were not observed in the non-PSTE group, an association between insomnia and disease activity cannot be ruled out. Further studies are needed to clarify this point, and we are now conducting a prospective study. We amended our description of the results in lines 195-196 and 246-249, and added sentences to the Discussion (lines 310-318) in the revised manuscript.

6. The authors used CES-D as a continuous variable and showed the significant difference between remission and active patients in Group 1 in Figure 2. It may be better to use categorical variable such as depressive or not based on the cut-off written in the method section. If the authors use CES-D as a continuous variable, please discuss minimal clinically important difference (MCID) of CES-D.

<Answer> We re-analyzed the data using categorical variables with Fisher’s exact test, and showed that a higher proportion of active patients were in a depressive state than in a non-depressive state in the PSTE group, but this difference was not observed in the non-PSTE group. 

 Depressive state Non-depressive state p-value

Active disease, n(%) 

 PSTE group 97 (42.9) 96 (26.2) .0001

 Non-PSTE group 18 (34.0) 41 (29.3) 0.6001

As shown in Supplementary Figure 4 (new S3 Fig), however, the CES-D score correlated significantly with the clinical activity indices in the PSTE group, but not in the non-PSTE group, although both scores were used as continuous variables. Therefore, we would like to present the data as continuous variables. There is no defined MCID for the CES-D score, but, according to the reviewer’s suggestion, we added the median values and interquartile range (IQR) in the figure legend of new Fig 3 (former Figure 2) in the revised manuscript.

7. Instead of table 1 and table 2, it may be better to show only patient characteristics in all patients, Group 1, and Group 2 in the same table without any statistical testing. In other words, the authors may delete table 1 and table 2, and they create a new table showing the patient characteristics.

<Answer> In response to this comment, we now provide the patient characteristics in the new Supplementary Table 1, which combines the previous Table 1 and 2, and relabeled the previous Supplementary Table 1 as a new Table 1. We understand that showing all the factors in univariate analyses is not always necessary, but we also consider that the multivariate analyses shown in the old Table 1 and 2 are very important data. We therefore revised these tables by showing only the factors for which multivariate analyses were performed (new Table 2 and 3, respectively).

8. I believe that the supplementary figure 1 is important because it shows patients’ selection. Please transfer the supplementary figure 1 to figure 1 in the manuscript.

<Answer> We modified and moved Supplementary Figure 1 to Fig 1, as suggested by the reviewer in comment No.10.

9. There may be too many figures and supplementary figures. Please decrease the number of figures only focus on exacerbation factors and the subgroup analysis of Group 1/2 regarding depression and insomnia because this may be a main research question of this study. If this is not, the authors should show what is their main research question.

<Answer> During the review process, we decided that analyses of the detailed insomnia symptoms in the insomnia (+) and insomnia (-) groups did not need to be shown in the present form. We therefore deleted the previous Figure 4, Supplementary Figure 6, and related descriptions in the manuscript were modified. We also deleted previous Supplementary Figures 2 and 5 in which the proportions of patients are shown, and the related descriptions in the manuscript were amended. 

10. As shown in the supplementary figure 1, the authors selected the study population in two ways. If the authors decrease the number of figures and analyses, it may be possible to select patients in one process.

<Answer> As we performed the depression and insomnia analyses similarly, we amended the new Fig 1 (former Supplementary Figure 1) in which the study is shown as one process.

11. In the overall analyses, the authors showed only p-values in figures. Please show the point estimates such as mean difference and 95% intervals. This is important to discuss whether the difference is clinically important or not. This issue is related to MCID as I commented in No 6.

<Answer> For analyses with continuous variables by which the point estimates can be shown such as in Figure 2 (new Fig 3), Figure 5 (new Fig 4), and Supplementary Figure 3 (new S2 Fig), we show median values with the IQR described in the figure legends. For analyses with categorical variables such as in Figure 3 (new S4 Fig), we show the number and ratio of patients in each group. 

12. The authors should carefully interpret results in this study. Because this is the cross-sectional study, it is reasonable to consider the depression or insomnia is due to disease activity rather than vice-versa. Please discuss the results from the viewpoint above. Moreover, the authors should explain how patients’ belief (psychological stress triggered their disease exacerbation) affects the association of depression or insomnia with disease activity. In other word, please explain the mechanism of this effect modification if the authors find the significant interaction (please see the comments No 5).

<Answer> As described in the limitation section, we understand that this study design does not clarify whether a depressive state or insomnia are a cause or consequence of the exacerbated symptoms, and would like to emphasize that when patients are divided into PSTE and non-PSTE groups, different associations between the disease activity and depressive state/insomnia are observed in each group. The findings of an effect modification differed between depressive state and insomnia, indicating that a patient’s belief in psychologic stress-triggered disease exacerbation is strongly connected to the association between the depressive state and disease activity, although an association between insomnia and disease activity cannot be ruled out. A depressive state might be more strongly involved than insomnia in disease activity in patients recognizing psychologic stress-triggered disease exacerbation, but in the present study design, a direct comparison cannot be performed and a prospective study targeting this issue is needed. Thank you for the important suggestion and we have now added sentences addressing this point in lines 319-327 of the revised manuscript. 

13. Overall, there may be better words and sentences to show what the authors want to discuss in this manuscript. Please ask English editing to native and academic English editors.

<Answer> The manuscript was edited by professional native-English speaking science editors from SciTechEdit International, LLC. 

1. It is better to show the study design in detail in the title. This is a multicenter cross-sectional survey or multicenter cross-sectional study.

<Answer> We changed the title to indicate that this was a multicenter cross-sectional study. 

2. In the introduction, the authors do not need to discuss seasonality because this is different from the current study purpose.

<Answer> We deleted the sentences describing seasonality in the Introduction.

3. In the method section, please briefly mention the validity and reliability of CES-D and insomnia questionnaire based on previous questionnaire development study conducted by the developers.

<Answer> The CES-D scale was validated by Yokoyama E, et al. (Sleep. 2010. Reference No.21) As for the insomnia questionnaire, we used a simple version of the sleep questionnaire based on the validated Insomnia Severity Index (ISI) (Bastien CH, et al. Sleep Med. 2001, which has been added as Reference No. 23) to reduce recall bias. It is commonly given to patients to assess their present sleep conditions in clinical practice. We briefly described the questionnaire in lines 164-166 of the revised manuscript. 

4. In the introduction and discussion, the author should summarize previous studies evaluating patients’ belief of disease exacerbation triggers and discuss the discrepancy between the results of current study and those of previous studies with possible reasons.

<Answer> Patients’ beliefs about some environmental factors affecting IBD, such as diet (Limdi JK, et al. Inflamm Bowel Dis. 2016, which has been added as Reference No. 15) and smoking (Saadoune N, et al. Eur J Gastroenterol Hepatol. 2015, which has been added as Reference No. 16) have been reported, but no studies have focused on patients’ beliefs regarding psychologic stress and disease activity. The present data provide new insight that a depressive state is one of the factors associated with disease activity in a subgroup of IBD patients who believed that psychologic stress triggered an exacerbation of their disease. We now describe this in lines 91-93 in the Introduction and lines 282-285 in the Discussion of the revised manuscript.

5. In the introduction and discussion, the author should summarize previous studies evaluating depression or insomnia and disease activity, and please discuss the discrepancy between the results of current study and those of previous studies with possible reasons.

<Answer> As described in the Introduction, there are conflicting reports regarding whether psychologic state and insomnia are associated with disease activity. In the present study, we found that depressive state is a factor associated with disease activity in a subgroup of IBD patients who believed that psychologic stress triggered an exacerbation of their disease. Previous conflicting reports might be due to the lack of patient stratification based on patients’ belief. We now discuss this in lines 285-287 of the revised manuscript. 

 

Response to Reviewer #2: 

Thank you very much for reviewing our manuscript. We have revised the manuscript in response to all comments received.

1. the discussion section should be altered to more accurately reflect the cross-sectional nature of the data. As an example, the statement " .. and the results suggest that psychological stress can be an initiator of disease" (line 281) is simply not appropriate. Similarly, the statement "our present data also suggest that high vulnerability to psychologic stress is associated with disease flare in patients with insomnia" (line 273) is not supported by the data.

<Answer> Thank you for the suggestion. We agree and have deleted these statements to avoid misunderstanding.

2. Finally, their conclusion "In conclusion, our data revealed that psychologic stress and insomnia affect disease activity in IBD patients, especially..." (line 292) is wrong, since it is just as likely that disease activity affects psychological stress and insomnia.

<Answer> As we described in the limitation section, in the present study, we could not clarify whether psychologic stress and insomnia affect disease activity or vice versa. Therefore, we changed the statement to “worsened mental state correlates with disease activity in IBD patients, especially...” in lines 342-343 of the revised manuscript, and deleted the word “insomnia” because of negative results regarding an effect modification between the PSTE and non-PSTE groups. Thank you for your suggestion. 

 

Response to Journal Requirements:

<Answer> We amended the manuscript so as to meet PLOS ONE’s style requirements. 

<Answer> The written informed consent was waived by the ethics committees by giving the participants the opportunity to opt out. We described this on lines 125-126 of the revised manuscript. 

3. Please include additional information regarding the questionnaire regarding potential factors of disease exacerbation used in the study and ensure that you have provided sufficient details that others could replicate the analyses. For instance, if you developed this questionnaire as part of this study and it is not under a copyright more restrictive than CC-BY, please include a copy, in both the original language and English, as Supporting Information.

<Answer> According to the suggestion by the Reviewer #1, we described the detailed information of the questionnaire in English on the Methods of the revised manuscript. We have also included the original Japanese version as S1 Fig in Supporting Information. 

'This work was supported by a Grant-in-Aid from the Japan Society for the Promotion of Science (Grant No. 26460969).'

'The authors received no specific funding for this work.'

<Answer> We have removed the funding-related text from the manuscript and updated the Funding Statement section in the online submission form.

'The authors have declared that no competing interests exist.' 

We note that one or more of the authors are employed by a commercial company: Kinshukai Infusion clinic

<Answer> Kinshukai Infusion clinic is not a commercial company but a medical corporation. The author has declared that no competing interests exist.

<Answer> This work was supported by a Grant-in-Aid from the Japan Society for the Promotion of Science (Grant No. 26460969). The funder did not have any additional role in the study design, data collection and analysis, decision to publish, or preparation of the manuscript. We have now updated the Funding Information of the online submission system and added above statements in lines 359-362 in Author Contributions of the revised manuscript.

<Answer> We have read the journal's policy and declare that the authors have no competing interests which can alter our adherence to PLOS ONE policies on sharing data and materials. We have amended the Disclosure Statement on lines 373-374 of the revised manuscript, and the cover letter. 

6. Thank you for including your ethics statement: The ethics committees of all participating institutions approved the study protocol.

<Answer> The study protocol was approved by the ethics committee of Osaka University Hospital, the ethics committee of National Hospital Organization Osaka National Hospital, the ethics committee of Kinshukai Infusion Clinic, the ethics committee of Osaka Rosai Hospital, the ethics committee of NTT-West Osaka Hospital, the ethics committee of Sumitomo Hospital, the ethics committee of Toyonaka Municipal Hospital, the ethics committee of Otemae Hospital, the ethics committee of Sakai City Medical Center, the ethics committee of Itami City Hospital, the ethics committee of Nishinomiya Municipal Central Hospital, the ethics committee of Hyogo Prefectural Nishinomiya Hospital, the ethics committee of Osaka Police Hospital, the ethics committee of Yao Municipal Hospital, the ethics committee of National Hospital Organization Osaka-minami National Hospital, and the ethics committee of Higashiosaka City Medical Center. We described this on lines 114-125 of the revised manuscript and added the same text to the “Ethics Statement” field of the submission form.

<Answer> We included captions for Supporting Information files at the end of the manuscript, and updated in-text citations.

---

## [Decision Letter · Decision Letter 1]

14 Apr 2020

PONE-D-19-26952R1

Psychologic stress and disease activity in patients with inflammatory bowel disease: A multicenter cross-sectional study

PLOS ONE

Dear Dr. Iijima,

Thank you for submitting your manuscript to PLOS ONE. After careful consideration, we feel that it has merit but does not fully meet PLOS ONE’s publication criteria as it currently stands. Therefore, we invite you to submit a revised version of the manuscript that addresses the points raised during the review process.

Again the reviewer addressed several major concerns about your revised manuscript. Please revise your manuscript carefully.

We would appreciate receiving your revised manuscript by May 29 2020 11:59PM. To enhance the reproducibility of your results, we recommend that if applicable you deposit your laboratory protocols in protocols.io, where a protocol can be assigned its own identifier (DOI) such that it can be cited independently in the future. For instructions see: http://journals.plos.org/plosone/s/submission-guidelines#loc-laboratory-protocols

We look forward to receiving your revised manuscript.

Kind regards,

Kenji Hashimoto, PhD

Academic Editor

PLOS ONE

Reviewers' comments:

Reviewer's Responses to Questions

**Comments to the Author**

1. If the authors have adequately addressed your comments raised in a previous round of review and you feel that this manuscript is now acceptable for publication, you may indicate that here to bypass the “Comments to the Author” section, enter your conflict of interest statement in the “Confidential to Editor” section, and submit your "Accept" recommendation.

Reviewer #1: (No Response)

2. Is the manuscript technically sound, and do the data support the conclusions?

Reviewer #1: Partly

3. Has the statistical analysis been performed appropriately and rigorously? 

Reviewer #1: Yes

4. Have the authors made all data underlying the findings in their manuscript fully available?

Reviewer #1: No

5. Is the manuscript presented in an intelligible fashion and written in standard English?

Reviewer #1: Yes

6. Review Comments to the Author

Reviewer #1: This revision substantially improved the quality of reporting. However, there are several issues which the authors should clarify. Especially, the authors should explain why they believe that there is effect modification between PSTE groups (i.e. biological mechanism).

1. Please report the method to estimate P values for interactions in the Method section.

2. Please explain the possible biological mechanism which causes effect modification (i.e. interaction).

3. The authors answered that there is no defined MCID for CES-D. Please mention the search method for this (i.e. database for searching and search terms). I briefly searched and found one article related to MCID (PMID: 27300327). There should be more articles related to this issue.

4. In the results section of the abstract, please report the details of the results such as proportion, point estimate, and 95% CI.

5. Figure 1 is difficult to follow. Moreover, the numbers written in the Patients section seem different from those in Figure 1. Please check them. If the numbers are correct, please clearly report the study flow. The authors should also report how many patients visited the study sites during the study period and how many were actually recruited.

6. In the statistical analysis section, the description is too simple. Please report the details such as how they selected covariates for the logistic regression model. If the authors want to conduct causal inference, they need to select covariates based on theories instead of statistical covariate selection.

7. In figure 3, the authors compared between remission and active patients with stratification of PSTE. To harmonize the way of showing results to figure 3, figure 4 should also show the comparison between remission and active patients with stratification of PSTE without subgrouping CD/UC.

8. The authors should rethink the reasons to conduct multivariable analyses. They excluded “problems with work or family” from covariates because they thought that this was a potential confounder. If this is a confounder, it is better to adjust it to evaluate the independent association. However, the authors excluded it. Please explain for what purpose they conducted multivariable analyses.

9. In Table 2 and Table 3, it is difficult to understand the results. Please use same method to conduct univariate and multivariable analyses such as logistic regression models. Moreover, the method of covariate selection and purpose of these analyses are unclear.

7. PLOS authors have the option to publish the peer review history of their article (what does this mean?). If published, this will include your full peer review and any attached files.

Reviewer #1: No

---

## [Author Response · Author response to Decision Letter 1]

2 May 2020

Response to Reviewer #1: 

Thank you very much for reviewing our manuscript. We have now revised the manuscript in response to all comments received. 

1. Please report the method to estimate P values for interactions in the Method section.

<Answer> To estimate P values for interactions, two-way analysis of variance was used for continuous variables and Cox regression analysis for categorical data. We have added the information in the Method section.

2. Please explain the possible biological mechanism which causes effect modification (i.e. interaction).

<Answer> In the current epidemiological study design, we cannot elucidate the definite biological mechanism underlying the effect modification. Multivariate analysis showed that patients with the PSTE group are predominantly female and do not think that other environmental factors exacerbated disease (Table 2). As described in the Discussion, psychologic stress can influence gut inflammation through various mechanisms via the hypothalamus–pituitary–adrenal axis and the autonomic nervous system, resulting in the production of proinflammatory cytokines, activation of macrophages, and alteration of intestinal permeability and gut microbiota [26-28]. Previous reports also show that female sex is associated with anxiety in IBD patients [30]. Increased psychologic stress in the PSTE group may augment the fear for exacerbation and thereby affect these mechanisms, whereas other factor(s) may affect disease activity in the non-PSTE group. We are currently conducting a basic research to explore the pathophysiological role of psychologic stress for intestinal inflammation. We have added the discussion in the Discussion of the revised manuscript.

3. The authors answered that there is no defined MCID for CES-D. Please mention the search method for this (i.e. database for searching and search terms). I briefly searched and found one article related to MCID (PMID: 27300327). There should be more articles related to this issue.

<Answer> In the paper presented by the Reviewer, the MCID for 15-item form of CES-D scale was investigated for German patients with depression, which is different from the 11-item CES-D we have used in the present manuscript. We searched papers using PubMed/MEDLINE with the search query of “11-item CES-D”[All Fields] AND “MCID”[All Fields] and no paper was found. We have additionally tried to search papers with the query of "MCID"[All Fields] AND "CES-D"[All Fields], three papers were found. MCID shown in the first paper was what the Reviewer #1 presented, the second was what the authors empirically considered for 20-item CES-D (PMID: 31037211), and third was not for CES-D (PMID: 31562683), so none of them can account for our query. Moreover, previous reports using CES-D score were targeting patients who had already been diagnosed as depression, which is totally different from our target of patients who are not diagnosed as depression in almost all cases. In the present study we would just like to investigate the statistically significant difference in the CES-D scores between the PSTE group and the non-PSTE group, and would not like to observe the decrease in the CES-D score by clinical intervention. 

4. In the results section of the abstract, please report the details of the results such as proportion, point estimate, and 95% CI.

<Answer> We have added the details of the results within the word count limit in the revised abstract.

5. Figure 1 is difficult to follow. Moreover, the numbers written in the Patients section seem different from those in Figure 1. Please check them. If the numbers are correct, please clearly report the study flow. The authors should also report how many patients visited the study sites during the study period and how many were actually recruited.

<Answer> In the first draft, we displayed the depressive state and insomnia analysis charts separately. In the second submission we revised as the Reviewer #1 instructed us to combine them, which seemed to be harder to follow the study flow. We have now renewed Figure 1 and clarified the study flow in the manuscript. And we actually recruited a total of 1078 cases, and the number of the patients who visited the study sites is unknown in this study design. 

6. In the statistical analysis section, the description is too simple. Please report the details such as how they selected covariates for the logistic regression model. If the authors want to conduct causal inference, they need to select covariates based on theories instead of statistical covariate selection.

<Answer> In logistic regression analysis, the covariates with significant difference in the univariate analysis were selected. We have added the explanation to the Statistical analysis section.

7. In figure 3, the authors compared between remission and active patients with stratification of PSTE. To harmonize the way of showing results to figure 3, figure 4 should also show the comparison between remission and active patients with stratification of PSTE without subgrouping CD/UC.

<Answer> In Figure 3 the factors related to PSTE are analyzed, while in Figure 4 the factors related to insomnia are analyzed, both of which are classified into the PSTE group and the non-PSTE group. Unlike CES-D, the insomnia score is a binary value, and if CD and UC are to be analyzed without distinction, disease activity must be expressed as a binary value in the active phase and the remission phase because the disease activity scores are different between CD and UC. Therefore, we have amended Figure 4 as the active patient group and the remission patient group were set on the horizontal axis as in Figure 3, and the vertical axis was defined as the proportion of insomnia patients. Analysis by Pearson’s chi-square test showed that, in the PSTE group, the proportion of insomnia patients was significantly higher in active patients than in those in remission, but the difference was not observed in the non-PSTE group. The P value of the interaction was 0.437, and no interaction was observed in both groups. These results are same as the former analysis, but thanks to the Reviewer’s suggestion, we have now clearly showed that insomnia is associated with disease activity especially in patients with the PSTE group.

8. The authors should rethink the reasons to conduct multivariable analyses. They excluded “problems with work or family” from covariates because they thought that this was a potential confounder. If this is a confounder, it is better to adjust it to evaluate the independent association. However, the authors excluded it. Please explain for what purpose they conducted multivariable analyses.

<Answer> In the first analysis, the alternative “problems with work or family” was excluded from the analysis because it seemed to be strongly related to psychologic stress and the results might be confusing to the readers. When we actually put it in variables, problems with work and family was shown to be an independent factor positively associated with PSTE, and other variables with significant difference in univariate analysis were remained as independent factors. These results indicate that patients in the PSTE group think that problems with work or family exacerbate the disease but other environmental factors do not. We have now amended Table 2. 

9. In Table 2 and Table 3, it is difficult to understand the results. Please use same method to conduct univariate and multivariable analyses such as logistic regression models. Moreover, the method of covariate selection and purpose of these analyses are unclear.

<Answer> Table 2 analyzes factors related to PSTE, and Table 3 analyzes factors related to insomnia. In the first draft of Table 2 and 3, the results of the univariate analysis for all the variables were shown, but we deleted the variables without significant difference in the univariate analysis according to the Reviewer #1’s suggestion. The analysis methods are the same. However, with the change in Figure 4 for the Question No.7, we would like to emphasize psychologic stress rather than insomnia, and have now deleted the Table 3 to avoid confusing.

---

## [Editor Report · Decision Letter 2]

5 May 2020

Psychologic stress and disease activity in patients with inflammatory bowel disease: A multicenter cross-sectional study

PONE-D-19-26952R2

Dear Dr. Iijima,

We are pleased to inform you that your manuscript has been judged scientifically suitable for publication and will be formally accepted for publication once it complies with all outstanding technical requirements.

With kind regards,

Kenji Hashimoto, PhD

Section Editor

PLOS ONE
---

## [Editor Report · Acceptance letter]

14 May 2020

PONE-D-19-26952R2 

Psychologic stress and disease activity in patients with inflammatory bowel disease: A multicenter cross-sectional study 

Dear Dr. Iijima:

I am pleased to inform you that your manuscript has been deemed suitable for publication in PLOS ONE. Congratulations! Your manuscript is now with our production department. 

With kind regards,

on behalf of

Prof. Kenji Hashimoto 

Section Editor

PLOS ONE